# Assessing Landslide Susceptibility along India's National Highway 58: A Comprehensive Approach Integrating Remote Sensing, GIS, and Logistic Regression Analysis

Mukta Sharma [1] , Ritambhara K. Upadhyay [2] , Gaurav Tripathi [3] , Naval Kishore [2], Achala Shakya [4] , Gowhar Meraj [5,*] , Shruti Kanga [6] , Suraj Kumar Singh [3] , Pankaj Kumar [7] , Brian Alan Johnson [7] and Som Nath Thakur [8]

[1]  School of Built Environment, I.K. Gujral Punjab Technical University, Jalandhar 144603, Punjab, India; mukta1087@ptu.ac.in
[2]  Department of Geology, Panjab University, Chandigarh 160014, India; ritambharaku@gmail.com (R.K.U.); navalkishorepu@gmail.com (N.K.)
[3]  Centre for Climate Change and Water Research, Suresh Gyan Vihar University, Jaipur 302017, India; gaurav.tripathi@mygyanvihar.com (G.T.); suraj.kumar@mygyanvihar.com (S.K.S.)
[4]  School of Computer Sciences, University of Petroleum and Energy Studies, Dehradun 248007, India; achala.shakya@ddn.upes.ac.in
[5]  Department of Ecosystem Studies, Graduate School of Agricultural and Life Sciences, The University of Tokyo, Tokyo 113-8654, Japan
[6]  Department of Geography, School of Environment and Earth Sciences, Central University of Punjab, VPO-Ghudda, Bathinda 151401, India; shruti.kanga@cup.edu.in
[7]  Institute for Global Environmental Strategies, Hayama 240-0115, Japan; kumar@iges.or.jp (P.K.); johnson@iges.or.jp (B.A.J.)
[8]  Department of Geography, Panjab University, Chandigarh 160014, India; thakur.geog@gmail.com
*  Correspondence: gowharmeraj@g.ecc.u-tokyo.ac.jp or gowharmeraj@gmail.com

**Abstract:** The NH 58 area in India has been experiencing an increase in landslide occurrences, posing significant threats to local communities, infrastructure, and the environment. The growing need to identify areas prone to landslides for effective disaster risk management, land use planning, and infrastructure development has led to the increased adoption of advanced geospatial technologies and statistical methods. In this context, this research article presents an in-depth analysis aimed at developing a landslide susceptibility zonation (LSZ) map for the NH 58 area using remote sensing, GIS, and logistic regression analysis. The study incorporates multiple geo-environmental factors for analysis, such as slope aspect, curvature, drainage density, elevation, fault distance, flow accumulation, geology, geomorphology, land use land cover (LULC), road distance, and slope angle. Utilizing 50% of the landslide inventory data, the logistic regression model was trained to determine correlations between causal factors and landslide occurrences. The logistic regression model was then employed to calculate landslide probabilities for each mapping unit within the NH 58 area, which were subsequently classified into relative susceptibility zones using a statistical class break technique. The model's accuracy was verified through ROC curve analysis, resulting in a 92% accuracy rate. The LSZ map highlights areas near road cut slopes as highly susceptible to landslides, providing crucial information for land use planning and management to reduce landslide risk in the NH 58 area. The study's findings are beneficial for policymakers, planners, and other stakeholders involved in regional disaster risk management. This research offers a comprehensive analysis of landslide-influencing factors in the NH 58 area and introduces an LSZ map as a valuable tool for managing and mitigating landslide risks. The map also serves as a critical reference for future research and contributes to the broader understanding of landslide susceptibility in the region.

**Keywords:** landslide; landslide susceptibility zonation; logistic regression; LULC; GIS

## 1. Introduction

India is among the top three countries globally (along with the USA and China) in terms of the number of natural disasters experienced in recent years [1]. According to the Geological Survey of India (GSI), approximately 15% of the Indian landmass is susceptible to landslides. The Himalayan region is especially susceptibility to landslides, particularly during the monsoon season, and these landslides can be caused by a variety of factors such as steep slope angles, fragile slope material, peculiar structural discontinuities, large-scale anthropogenic activities, and sudden weather events such as cloud bursts. These factors can combine to create a hazardous environment for the communities living in the region. In light of the significance of the Himalayan region, there has been significant research into landslides within the NH 58 area. Noteworthy studies include the work of Veerappan et al. (2017), which utilized frequency ratio and analytical hierarchy process models [2], Saha et al. (2021), who employed ensemble methods like conditional probability and boosted regression trees [3], Malik et al. (2016), who introduced the "landslide nominal risk factor" [4], and Guri et al. (2015), who applied weights of evidence modeling [5]. While these studies have made important contributions, methodological gaps and areas where a fresh analytical perspective is needed remain.

The recent past has seen substantial infrastructure development in the Himalaya region, marked by the construction of numerous dams, reservoirs, tunnels, road networks, hydroelectric power plants, and townships. While the upside of such development work is undeniable, major challenges related to the sustainability and geo-environmental safety of this infrastructure have also emerged. A plethora of studies corroborate the heightened incidence of slope failures and land subsidence following such developmental projects [6,7]. For instance, persistent scatterer interferometry (PSI) has been employed to monitor the impact of reservoirs on slope stability, notably around the Baglihar Dam Reservoir and Tehri region [6,8]. The analyses indicate that both reservoir drawdown effects and localized rainfall patterns contribute to land deformations, with velocities ranging from −95.1 mm/year to +85.1 mm/year [9].

In light of these geo-environmental risks, the current study aims to develop a landslide susceptibility zonation (LSZ) map for an area along the National Highway (NH) 58 in India's Himalaya region, employing remote sensing, GIS, and logistic regression analysis. The NH 58 stretch connecting Rishikesh and Srinagar in Uttarakhand serves as a crucial corridor for tourism, trade, and strategic military movements, making its integrity vitally important for the state and the nation. This study will thus shed light on the contributing factors to landslides along this critical roadway and identify high-risk zones. The road connects major pilgrimage sites and is particularly vulnerable during the monsoon season, which sees frequent slope failures—often in road cuts that are left untreated. Materials such as shale, phyllite, quartzite, soil, and debris that form the slopes along NH 58 are known to be inherently vulnerable to slope failure, especially when disturbed for infrastructure development. Therefore, the present study aims to address these vulnerabilities, which are further exacerbated by factors like the absence of pre-construction geological and geomorphological assessments [7].

Landslide susceptibility zonation (LSZ) mapping is a preliminary step towards landslide hazard mitigation [8–12]. The practice of LSZ mapping involves the identification of zones with a higher probability of experiencing landslides, based on an analysis of various geo-environmental factors [13–19]. LSZ is based on the principle that the set of conditions responsible for past and present landslides will likely also help to predict future landslide occurrences. In accordance with Brabb's definition of landslide susceptibility mapping, LSZ aims to determine the spatial likelihood of landslides occurring at a particular geographic location, based on a range of geo-environmental factors. LSZ mapping is typically carried out using heuristic methods [20–22], probabilistic/statistical methods [23–29], semi-quantitative/logical approaches [30–37], deterministic or physically based methods [38–42], and machine learning methods [43–48]. Heuristic methods for LSZ mapping rely on the expertise and experience of professionals in assessing the field conditions. Based on field

assessments, professionals assign relative weights to the landslide-causing factors and integrate them to generate an LSZ map. Most of the statistical methods for LSZ mapping come under probabilistic frameworks. These approaches assess the landslide susceptibility in probabilistic terms, e.g., using the weight of evidence, logistic regression, or information value. Semi-quantitative/logical methods leverage the potential of logical tools like the AHP (analytical hierarchy process), fuzzy logic approach, combined landslide frequency ratio and fuzzy logic, and weighted linear combination (WLC) to combine the weights of the factors. Deterministic or physically based approaches are mainly suited to site-specific conditions. In these methods, geotechnical factors including the state of structural discontinuities, the strength of the material that forms the slope, the water pressure in the pores, the cohesion of the slope material, and the shape of the slope are analyzed and modeled to assess the stability of the slope faces. As computing technology has advanced, statistical methods have progressed into machine learning techniques [49]. Based on the landslide distribution within the factors, machine learning methods use rigorous computation to estimate the landslide hazard zones. Pros and cons of each method can be found in the literature.

The study utilized logistic regression to forecast the likelihood of landslides on the NH 58 stretch connecting Rishikesh and Srinagar in Uttarakhand. Numerous statistical approaches, including linear regression, discriminant analysis, and logistic regression, have been investigated in the literature to identify areas prone to landslides. More specifically, the logistic regression method has been used in a variety of terrain conditions due to its advantage over linear regression and the discriminant method in terms of accommodation of continuous as well as categorical raster data. The logistic regression method has proven to be advantageous in landslide susceptibility analysis due to its ability to exclude irrelevant factors from the model [50]. It has been widely used by various researchers in the Himalayan region, and its efficacy in predicting landslide occurrences has been observed. The LR model not only characterizes the chosen factors but also computes the relative contribution of different classes towards landslide occurrence. Additionally, insignificant classes can be removed from the analysis, and probability can be estimated on a grid-by-grid basis using this method [51–53].

## 2. Study Area

The road stretch of NH 58 from Rishikesh to Srinagar in Uttarakhand, India, plays a crucial role in connecting the Pauri Garhwal region with the rest of the country. This section of NH 58 is an important link that contains pilgrimage sites and recreational activity locations. In the last 30 years, this region has witnessed prosperity and that has led to an unprecedented rise in traffic flow. Additionally, this NH is used by lakhs of tourists every year mainly between April and September. Altogether, it becomes a grave situation when any slope failure along the road section blocks traffic. Even though global climate change has impacted this region, it still experiences a relatively milder summer compared to other part of the Indian plains. The winter remains colder and there are areas which experience subzero temperatures during the winter. Chemical weathering phenomena are significantly accelerated due to the warmer days and cooler nights, regardless of seasonal variations [54]. During the monsoon season, the region (Figure 1) receives rainfall in the range of 1000 to 2000 mm (Source: Geological Survey of India), which causes number of slope failure instances along the entire stretch of the road section.

Over countless millennia, intricate geological activity has given rise to the Himalayan landscape. Nestled between the towering Greater Himalayas and the Siwalik ranges lie the Lesser Himalayan ranges, flanked on either side by the Main Central Thrust (MCT) and Main Boundary Thrust (MBT), respectively [54]. The Lesser Himalayan sequence incorporates a variety of rock types such as metasedimentary rocks, metavolcanic rocks, and augen gneiss [55–57]. The sequence is generally categorized into two distinct sections, termed as the inner and outer Lesser Himalayan sequences [58]. According to the study in [59], the outer Lesser Himalayas primarily contain formations like the Chakrata (mostly

made up of sandstone and siltstone), Rautgara (predominantly sandstone and quartzite), Mandali (comprising mainly slates and phyllite), Chandpur (largely phyllite), Nagthat (primarily quartzite), Blaini (early Proterozoic, mostly siltstone and slates), Krol (late Proterozoic, mostly limestone), Tal (early Cambrian, mostly sandstone), Bansi and Subathu (Cretaceous to Paleocene, mostly shelly limestone and sandstone), and Ramgarh group (mainly granitic, phyllite and siltstone). The Lesser Himalayan sequence is characterized by six major synclines: Garhwal, Nainital, Mussoorie, Pachmunda, Naglidhar, and Krol. The area under examination is located within the Garhwal syncline in the outer Lesser Himalayas, stretching along NH 58, from Rishikesh to Rudraprayag. A geological chart of the region is displayed in Figure 2.

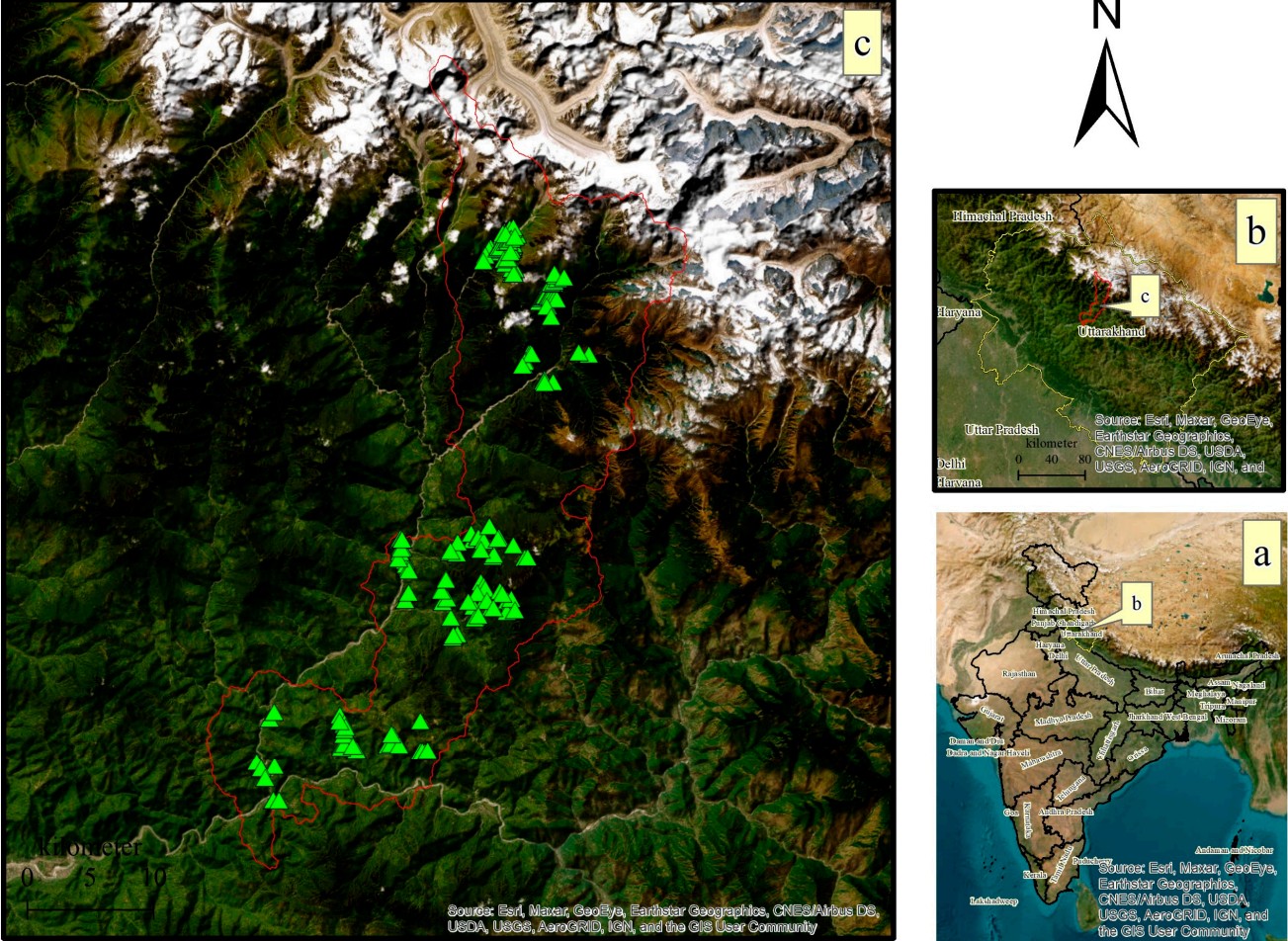

**Figure 1.** Map representing location of study area: (**a**) Union of India, (**b**) State of Uttarakhand, and (**c**) study area showing the landslide inventory used in this study to train the model. The green triangles are the locations of landslides that were used in this study.

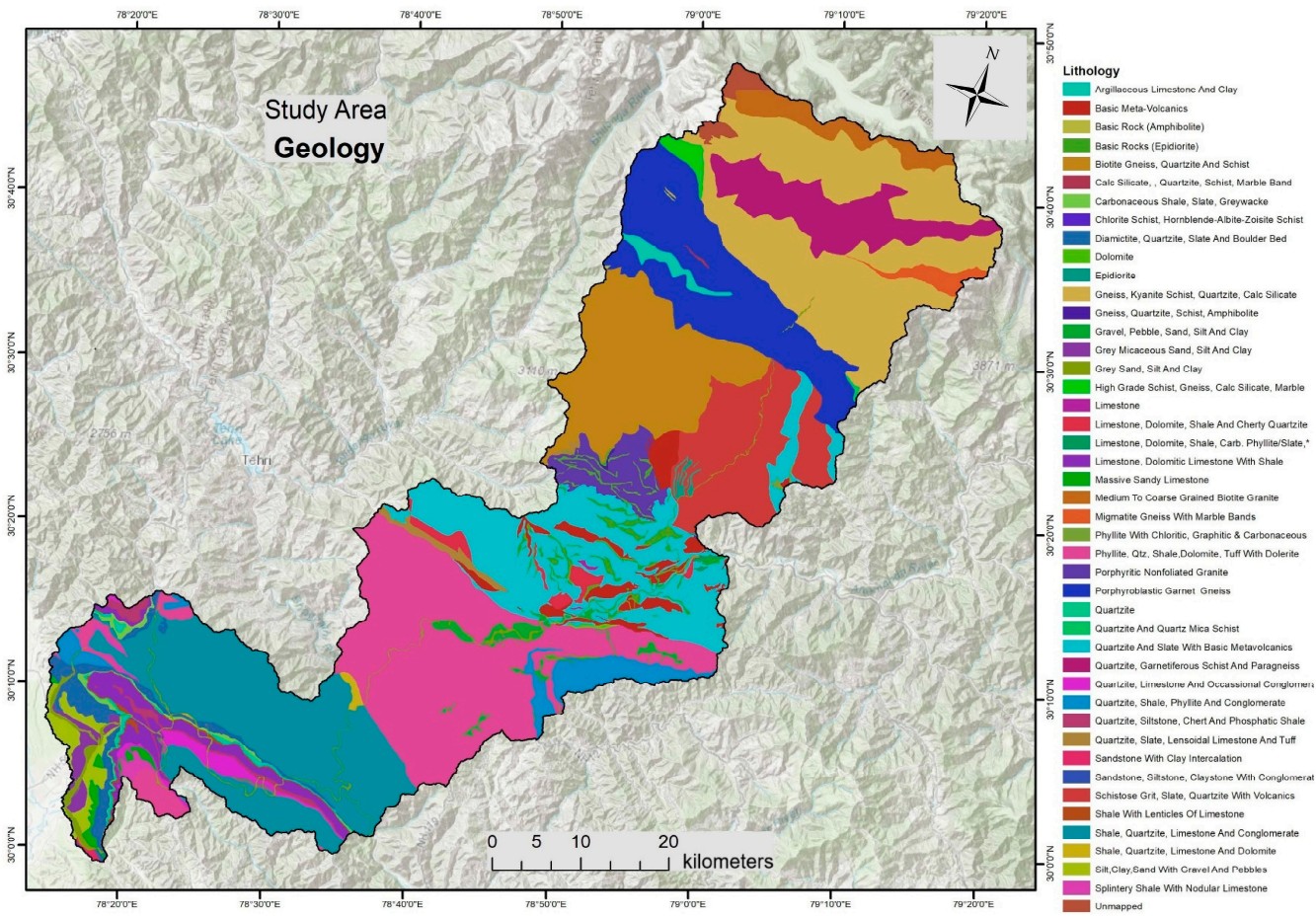

**Figure 2.** Map representing geology of study area.

## 3. Materials and Methods

### 3.1. Materials

The current study employs eleven factors, elevation, aspect, curvature, slope, flow accumulation, drainage density, land use/land cover (LULC), geology, distance from fault, geomorphology, and distance from road, to evaluate their contribution towards landslide susceptibility. The term landslide susceptibility denotes the likelihood or probability of the occurrence of landslides in a specific area based on the local terrain conditions [53]. ALOS (Advanced Land Observation Satellite) PALSAR DEM data were utilized to extract altitude, aspect, slope, flow accumulation, and curvature data layers with a 12.5 m spatial resolution in the ArcGIS 10.7 environment. QGIS was used to digitize the geomorphological map and geological fault map from the WMS layer from BHU-VAN. The Euclidean distance method was used in ArcMap 10.7 to derive the distance to the geological fault map. The geological map was obtained from BHUKOS (Geological Survey of India). Sentinel-2 imagery with a 10 m spatial resolution was classified using maximum likelihood classifier to generate a map of eight major land use and land cover classes (evergreen forest, deciduous forest, grassland, settlements, rocky barren, crop land, water bodies, and snow cover) using band 8 (NIR), band 4 (Red), and band 3 (Green). This imagery was processed in SNAP and Erdas Imagine 2014, and then verified using Google Earth Pro. Based on an accuracy assessment, we found that an overall accuracy of 78% was achieved for the LULC mapping. The road network was acquired from the OpenStreetMap (OSM) service layer in QGIS and consulted with Google Earth Pro. The Euclidean distance formula was used to determine the distance from roads. The landslide distribution map database was assembled using a myriad of resources, such as historical images from Google Earth, imagery from Landsat 4–5 TM, 7 ETM+, 8 OLI, LISS III, high-definition Sentinel-2, high-definition world imagery from

ArcMap, and extensive field study data. The final analysis in R studio utilized 2300 mapped and processed landslide locations. Further details concerning the extracted data layers are presented in the subsequent section [60,61]. The details of the data used in this study are shown in Table 1.

**Table 1.** Sources of data used in this study.

| S. No. | Data Type | Product Used | Source | Software/Platforms |
|--------|-----------|--------------|--------|--------------------|
| 1 | Digital Elevation Model (DEM) | Elevation, Aspect, Curvature, Slope, Flow Accumulation, Drainage Density | ALOS PALSAR DEM (12.5 m) | ArcGIS 10.7 |
| 2 | Multi-Spectral Data | Land Use and Land Cover | Sentinel-2 (10 m) | ERDAS Imagine 10.1 |
| 3 | Geology and Fault Lines | Geology and Distance from Fault | Geological Survey of India (1:50 k) | QGIS (12.4) |
| 4 | Geomorphology | Geomorphology | NRSC (1:50 k) | QGIS (12.4) |
| 5 | Road Network | Distance from Road | OpenStreetMap (1:1 k) | QGIS (12.4) |
| 6 | Training Data | Landslide Inventories | News Paper reports, Field observation | Google Earth Pro |

The slope aspect of a topographic surface refers to the direction that a slope faces relative to the north direction. It influences the amount of sunlight and rainfall that the slope receives, which in turn impacts temperature, soil moisture, and vegetation cover. In the study area, south-facing slopes receive more sunlight and rainfall, making them warm, wet, and thickly vegetated. Conversely, slopes facing north in this region are cold, arid, and covered in glaciers. For the purpose of landslide susceptibility analysis, this aspect was segregated into nine categories: flat (−1), north (both 0–22.5° and 337.5–360°), northeast (22.5–67.5°), east (67.5–112.5°), southeast (112.5–157.5°), south (157.5–202.5°), southwest (202.5–247.5°), west (247.5–292.5°), and northwest (292.5–337.5°) (Figure 3a).

Information regarding the slope angle was sourced from the ALOS-PALSAR DEM, where each grid symbolizes the angle of inclination relative to a flat surface (illustrated in Figure 3c). The angle of a slope is an essential parameter in the onset of landslides and is a fundamental consideration in analyzing instability in mountainous areas. As the angle of the slope intensifies, so does the shear stress of the materials constituting the slope, which could potentially result in instability and landslides [31,42,49,51]. Flow accumulation, a parameter signifying the water flow transition from convex to concave areas and its subsequent gathering, is vital in pinpointing potential landslide-prone regions. It is calculated by assessing the landmass that provides surface water to a location which serves as a collection point. The extent of pixels or area contributing to the runoff of a specific pixel is what constitutes flow accumulation [45]. This factor plays a significant role in the analysis of slope instability, as it pinpoints areas where water accumulates post-rainfall, suggesting potential landslide zones during seismic activity. The flow accumulation map, presented in Figure 3e, was employed in the procedure of mapping landslide susceptibility.

The distribution of land use and land cover (LULC) is a crucial factor that can contribute to landslides in hilly regions. The stability of slopes is heavily dependent on LULC practices in such areas. To identify the LULC distribution in the study area, satellite imagery and topographic maps were combined to generate nine different classes: snow cover, rocky barren land, stream, grassland, evergreen forest, deciduous forest, crop land, and built-up areas (as shown in Figure 3d). Most of the landslides were found to be associated with built-up areas, barren land classes, and crop land classes. The layer depicting landslide events was superimposed on the LULC layer, and information regarding LULC was extracted. Using the counting method, the distribution of landslides among all LULC categories was determined. Profile curvature is a measure of the slope gradient in the direction of dip, and it plays a significant role in regulating flow velocity on the slope, which can influence landslide triggers [52–54]. It is an essential factor in landslide susceptibility mapping. A

DEM is generally used to calculate profile curvature, which results in negative values indicating upwardly convex slopes, positive values indicating upwardly concave slopes, and a value of zero indicating a linear surface [62,63]. Profile curvature influences the pace of flow, either speeding it up or slowing it down, across the face of a slope and is linked with processes of erosion and mass movement. A map highlighting profile curvature was developed, marking both concave and convex shapes (Figure 3b).

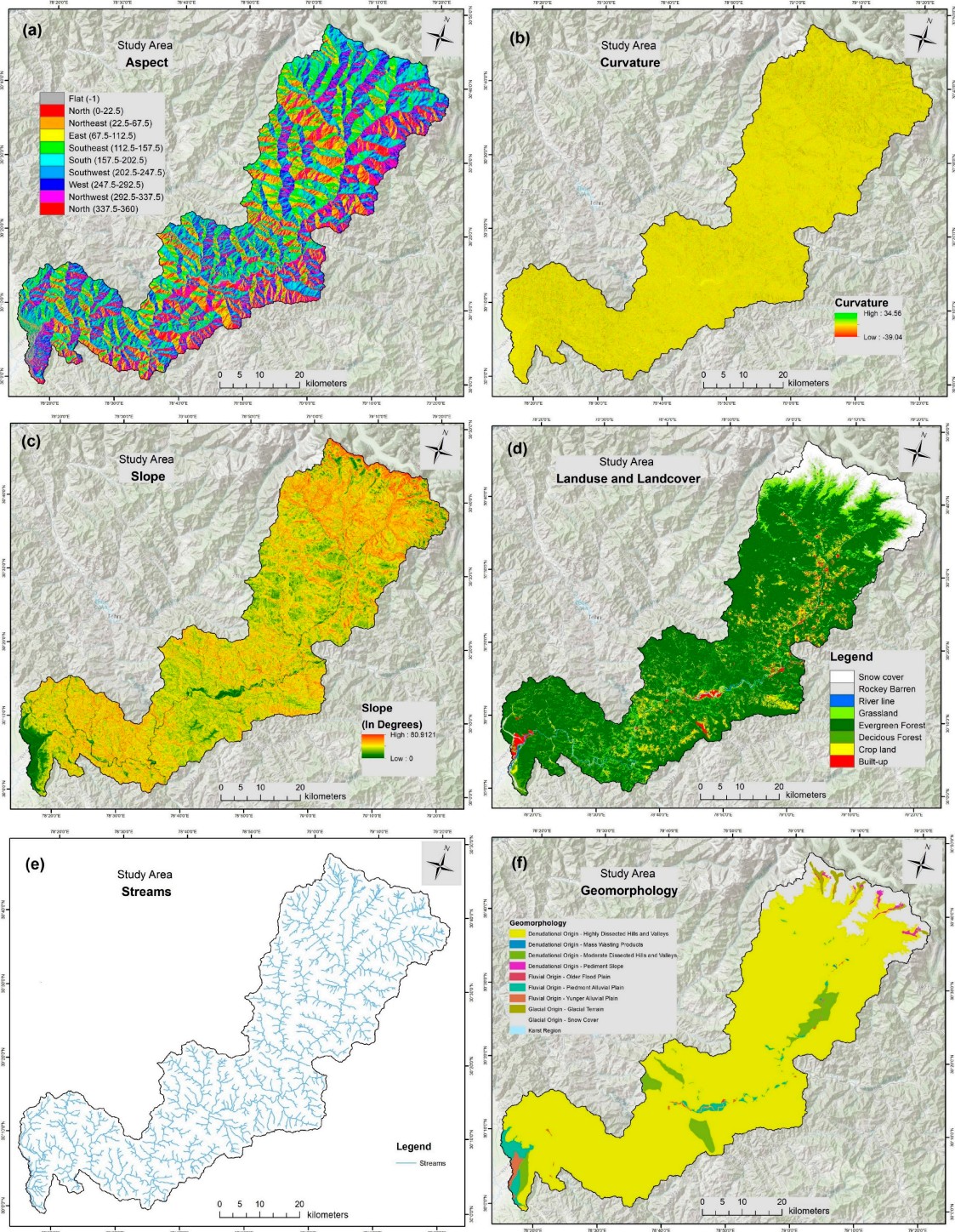

**Figure 3.** (**a**) Topographic aspect map, (**b**) slope curvature map, (**c**) topographic slope angle map, (**d**) land use and land cover map, (**e**) drainage map, and (**f**) map representing geomorphological features of the study area.

Drainage density refers to the total length of all the streams within a given drainage basin divided by the area of that drainage basin. It provides an estimation of the extent to which a watershed is drained by streams. In mountainous regions with undulating topography, high drainage density is common due to a higher bifurcation ratio, which can increase the risk of inundation. Areas with high drainage density are associated with higher occurrences of landslides, while a low drainage density reflects lower chances of landslides. The drainage density map for the study area can be found in Figure 3e. Geomorphic processes are central to the changing landscapes through time, including landslides. In this study, geomorphic features are interpreted from multi-source satellite images. The most dominant landform was found to be of denudational origin, of which the highly dissected hills and valleys category is most dominant. Mass wasting products, moderate dissected hills and valleys, and pediment slopes are other landform categories that are observed in the area. The landforms in the study area have been interpreted to have fluvial origins, including the older flood plain, piedmont alluvial plain, and younger alluvial plain. The geomorphological map in Figure 3f was created using multi-source satellite data to identify and classify these landforms.

The study area is situated on tectonically active Himalayan terrain. Geological discontinuities such as folds, faults, lineaments, shear zones, anticlines, synclines, etc., are commonly seen in this area. Research has indicated that the occurrence of landslides is associated with the presence of structural inconsistencies [64,65]. Linear geological discontinuities are commonly recorded as photo-lineaments using satellite imageries and DEMs. Field investigations have reflected that the frequency of landslides decreases with a decrease in proximity to lineaments. According to the field evidence, a lineament buffer layer was generated and used in the logistic regression (LR) model. In the Himalayas, a vast number of landslides are reported along the road section. Hill roads are constructed on the cut slopes, which are often left untreated, and thus susceptible to landslides. Within the study location, it was observed that the cut slopes remained untreated and were situated at a highly steep angle. Such untreated road-cut slopes frequently fail during the monsoon. Keeping this in mind, first a road network layer, then a buffer layer was prepared for LSZ mapping.

*3.2. Methods*

Logistic regression is a statistical modeling technique used to predict the likelihood or probability of a binary outcome, such as the occurrence or absence of landslides [55]. In the context of LSZ mapping, it can be used to evaluate the relationship between the incidence of landslides (dependent variable) and a group of independent variables, such as slope angle, aspect, curvature, lithology, and land use. By analyzing the relationship between these variables, it is possible to develop a model that can be used to predict the probability of future landslides in a given area, which is critical for developing effective hazard management strategies [66–70]. The logistic regression model estimates the probability of landslides occurring based on the values of the independent variables. This probability is calculated using a mathematical formula that takes into account the values of the independent variables and their coefficients, which are estimated from the data. The resulting model can be used to predict the likelihood of landslides occurring in different locations based on their respective values for the independent variables. The model can also be used to identify the most significant factors contributing to the likelihood of landslides, allowing for the development of effective landslide susceptibility maps and hazard management strategies:

$$p = 1/(1 + e^{-z}) \tag{1}$$

In logistic regression, the probability of landslides occurring, denoted by p, is estimated using a mathematical formula that involves a weighted linear combination of the independent variables. This weighted linear combination is represented by z and is calculated by multiplying the values of each independent variable by their respective coefficients and summing the products. The resulting value of z is then used in the logistic regression

equation to estimate the probability of landslides occurring. This equation is based on a sigmoidal function that maps the range of z values to a probability value between 0 and 1, representing the likelihood of landslides occurring for a given set of independent variables.

$$z = \log it(p) = Ln(p/(1-p)) = \beta_0 + \beta_1 X_1 + \cdots + \beta_n X_n \tag{2}$$

Logistic regression is a method that estimates the likelihood of landslides, represented by p, through a mathematical equation comprising a constant term ($C_0$) and coefficients ($\beta_1, \beta_2, \ldots, \beta_n$) for every independent variable ($X_1, X_2, \ldots, X_n$). The calculated landslide probability is then converted into odds or a likelihood ratio, depicted as $p/1-p$, which conveys the likelihood of landslides happening relative to them not happening. As the weighted linear combination of independent variables, denoted by z, fluctuates from negative to positive infinity, the landslide probability (p) shifts from 0 to 1 along an S-shaped curve, known as the sigmoid function. This curve illustrates the relationship between the independent variables and the landslide occurrence probability, providing a basis for developing effective models for landslide susceptibility mapping and hazard management. As a method, logistic regression is highly beneficial for predicting event probabilities, such as landslides, based on multiple independent variables. The predicted landslide probability for a raster point ranges from 0 to 1, where a value nearing 1 signifies a high landslide probability and a value approaching 0 implies a low landslide probability. A major benefit of logistic regression is that it does not require a linear relationship between independent and dependent variables, and it imposes fewer restrictions regarding the statistical assumptions necessary for the analysis. Another advantage is the flexibility in the measurement of independent variables, such as nominal, ordinal, interval, or ratio scales. For the logistic regression's application in landslide susceptibility mapping, this study used a dataset with 2300 landslide points and an equal quantity of non-landslide points. Independent variables were split into two categories: those related to the presence of landslides were given a value of 1, while the others were assigned a value of 0. This dataset was imported into the R programming environment and adapted to a logistic regression model using the generalized linear model (glm) function with a binomial family. The coefficients for the independent variables were computed, and the model can be used to forecast landslide likelihood in different regions based on these independent variable values. Instead of utilizing dummy variables, the study processed all continuous and categorical data as a single entity.

## 4. Result and Discussion

In the analysis of our logistic regression model, the results from the 'accuracy percentage prediction test' were elucidated. Upon creating a classification graph of 0 and 1 for both the observed and predicted groups, distinct patterns emerged. The landslide occurrence probability was assessed for all raster points. After cross-validation, distinct differences in the consistency of the hazard map [10] were observed. Following the division of the dataset into training and testing segments, the results indicated notable variations in model performance. The model's efficacy was ascertained through the values of McFadden's pseudo R-squared ($R^2$), Cox and Snell's maximum likelihood pseudo R-squared ($R^{2ML}$), and Nagelkerke's Cragg and Uhler's pseudo R-squared ($R^{2CU}$). Notably, a majority of the landslide raster points demonstrated a probability higher than 0.5, and vice versa for non-landslide points. The classification graph revealed distinct clusters for predicted probabilities, with successful models showing a stark contrast in the distribution of 0 points and 1 points.

$$\begin{aligned} Z = [&(-2.510) + \text{Elevation} \times (-5.616 \times 10^{-4}) + \text{Aspect} \times (-7.133 \times 10^{-4}) + \text{Curvature} \times (-5.533\text{-} \times 10^{-2}) \\ &+ \text{Geology} \times (7.984 \times 10^{-8}) + \text{Geomorphology} \times (1.756 \times 10^{-8}) + \text{Geological Fault} \times (-1.686 \times 10^{-5}) \\ &+ \text{LULC} \times (-6.075 \times 10^{-8}) + \text{Distance from Road} \times (-1.186 \times 10^{-3}) + \text{Flow Accumulation} \\ &\times (-2.714 \times 10^{-8}) + \text{Drainage Density} \times (1.104) + \text{Slope} \times (1.236 \times 10^{-1})] \end{aligned} \tag{3}$$

The logistic regression statistics gave the constant/intercept and coefficients for the independent variables, as displayed in Table 2. A positive coefficient suggests that the independent variable augments the likelihood of a landslide, while negative values infer a negative correlation with landslide probability. Utilizing Eq. 2 and Eq. 3, the probability estimate for landslides across the entire study area was determined, producing probability values between 0 and 1. Following this, the probability map was segmented into five categories: very low, low, moderate, high, and very high susceptibility zones, employing Jenk's natural break classification method [22]. The derived landslide susceptibility zone (LSZ) map for the study area is shown in Figure 4. The significance of the independent variables in determining the degree of landslide susceptibility has been indicated by the coefficient values ($\beta_i$). Positive and negative values of $\beta_i$ have an impact on landslide probability, while insignificant independent variables do not yield $\beta_i$ values. The logistic regression analysis in this study has shown positive $\beta$ values for drainage density, slope angle, geomorphology, and geology variables.

**Table 2.** Generalized linear regression coefficients.

| Variables | Estimate | Std. Error | z Value | Pr(>|z|) |
|---|---|---|---|---|
| (Intercept) | $-2.37$ | $3.62 \times 10^{-1}$ | $-6.562$ | $5.31 \times 10^{-11}$ *** |
| Aspect | $-7.13 \times 10^{-4}$ | $4.88 \times 10^{-4}$ | $-1.461$ | 0.14406 |
| Curvature | $-5.53 \times 10^{-2}$ | $3.68 \times 10^{-2}$ | $-1.503$ | 0.13273 |
| Drainage Density | 1.10 | $3.78 \times 10^{-1}$ | 2.924 | 0.00346 ** |
| Elevation | $-5.62 \times 10^{-4}$ | $9.03 \times 10^{-5}$ | $-6.221$ | $4.95 \times 10^{-10}$ *** |
| Fault | $-1.69 \times 10^{-5}$ | $1.91 \times 10^{-5}$ | $-0.882$ | 0.37768 |
| Flow Accumulation | $-2.71 \times 10^{-8}$ | $4.04 \times 10^{-8}$ | $-0.672$ | 0.50153 |
| Geology | $7.98 \times 10^{-8}$ | $2.73 \times 10^{-8}$ | 2.92 | 0.00350 ** |
| Geomorphology | $1.76 \times 10^{-8}$ | $4.92 \times 10^{-9}$ | 3.568 | 0.00036 *** |
| LULC | $-6.08 \times 10^{-8}$ | $4.45 \times 10^{-9}$ | $-13.648$ | $<2 \times 10^{-16}$ *** |
| Road Distance | $-1.19 \times 10^{-3}$ | $8.62 \times 10^{-5}$ | $-13.761$ | $<2 \times 10^{-16}$ *** |
| Slope | $1.24 \times 10^{-1}$ | $4.18 \times 10^{-3}$ | 29.546 | $<2 \times 10^{-16}$ *** |

** $p \leq 0.01$; *** $p \leq 0.001$.

The geomorphological factors have displayed notably high positive coefficient values. Geologically, the study area is diverse, since it is represented by a range of rocks with varying weathering conditions. As per the field observations and previous studies, the most problematic of these are weathered quartzite, phyllite, weathered slates, siltstones, and weathered sandstones. In addition to that, many of the slopes and terraces are supported by debris and river-borne materials (RBM) that are highly prone to failure. Positive $\beta$ values are observed for the slope angle factor, and this can be clearly seen on the ground. The terrain in the study area is characterized by rough and complex features such as cliffs, escarpments, spurs, ridges, and steep slopes that generally have moderate to very high inclinations. Most of the landslides seen here are found in slope faces that are dipping at high angles. A positive $\beta$ value has also been estimated for the drainage density factor. Several streams originate from the headland areas and form a complex drainage network. Since slope failures are directly attributed to moisture conditions, the high drainage density of the region reflects the propensity of landslides. In addition to geology, slope angle, and drainage density, the geomorphology factor has also showed positive $\beta$ values. Landforms of the study area are rooted in denudational, glacial, fluvial, and Karst origins. Landforms in the region are manifested by ridges, spars, deeply dissected valleys, steeply inclined slope faces, river terraces, debris slopes, and geologically governed streams, and thus, they are inherently susceptible to landslides.

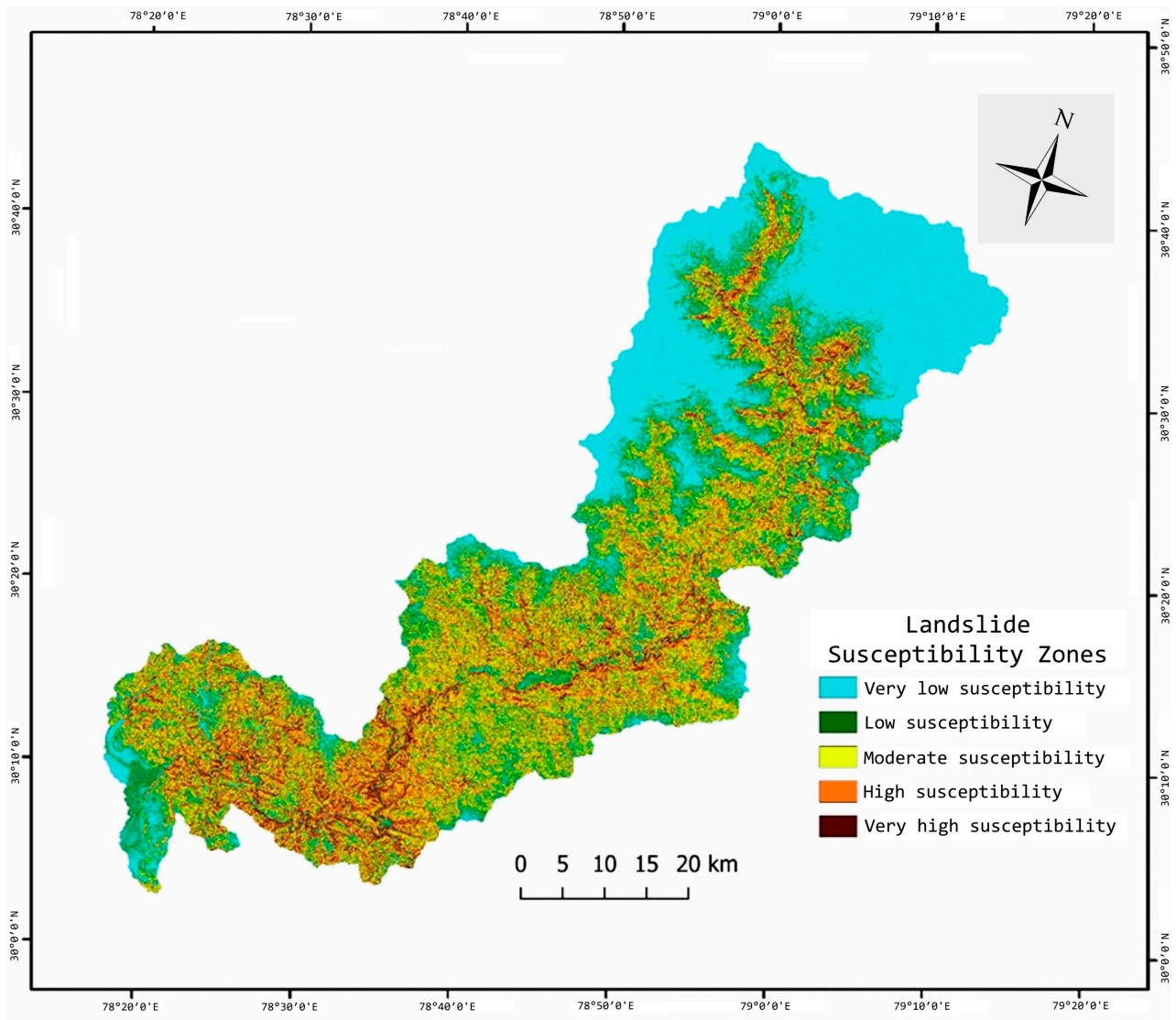

**Figure 4.** Landslide susceptibility map of the road along National Highway 58, India.

The verification of landslide susceptibility zone (LSZ) maps chiefly relies on the confusion matrix or contingency table (Table 3), which computes the intersecting areas between two binary maps, namely, the training and testing datasets. In the context of this study, continuous susceptibility maps were contrasted with the landslide inventory map to detect two kinds of errors: (1) landslides might transpire in areas projected to be stable, and (2) landslides might not happen in areas expected to be unstable. The receiver operating characteristic (ROC) curve method was employed for LSZ validation, with the findings illustrated in Figure 5. The ROC curve diagrams the model sensitivity (true positive fraction values), computed for varying threshold values, against the model specificity (true negative fraction values). The sensitivity ratio is the count of correctly classified presence data divided by all presence data, while the specificity ratio is the tally of correctly classified grid cells without landslides divided by all grid cells without landslides. The area beneath the ROC curve signifies the model's predictive accuracy, where a value of 1 denotes impeccable prediction, and a value around 0.5 implies model failure. In the context of this study, the ROC curve yielded a value of 0.920, denoting a predictive accuracy of 92%.

**Table 3.** Classification table and statistical performance of the model.

| Observed | Predicted | | Model Accuracy (%) |
|---|---|---|---|
| | Absence of Landslide (0) | Presence of Landslide (1) | |
| Absence of landslide (0) | 1988 | 312 | 86.4 |
| Presence of landslide (1) | 276 | 2024 | 88.0 |
| Overall accuracy (%) | | | 87.2 |

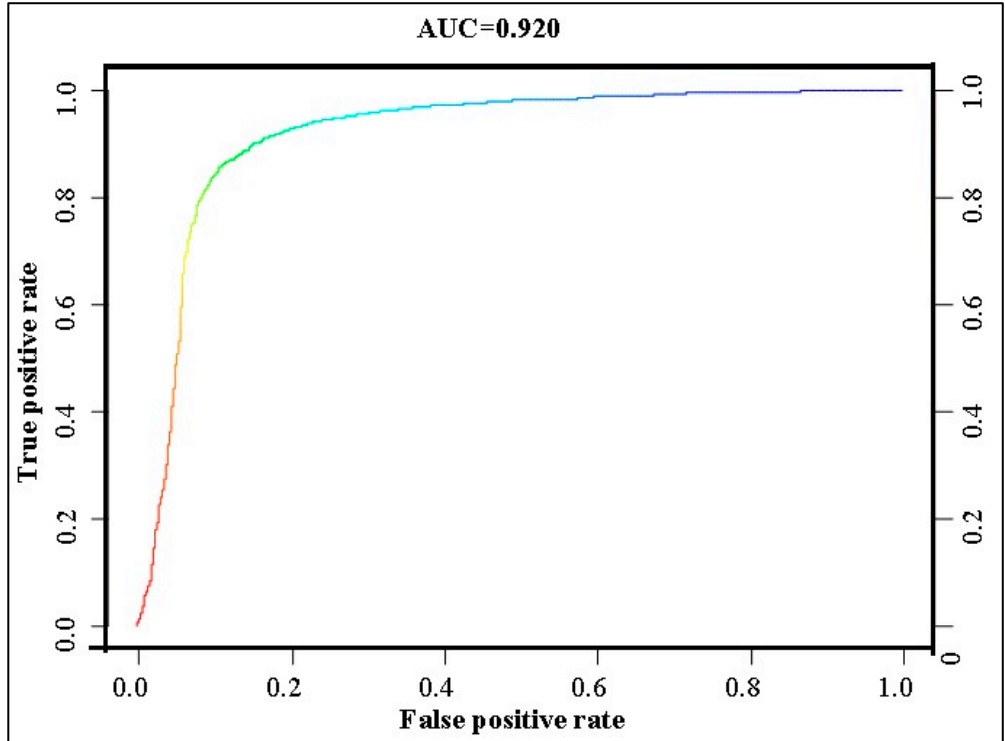

**Figure 5.** ROC curve reflecting model prediction rate.

The results of our study revealed distinct patterns in landslide susceptibility within the Himalayan region surrounding National Highway 58. Specifically, areas exhibiting high landslide susceptibility were predominantly found in the catchment's upper reaches. These areas were marked by their steep slopes and significant agricultural land use. On the other hand, the lower catchment reaches displayed low susceptibility, characterized by their gentler slopes and prevalent urban land use.

Using logistic regression for landslide susceptibility zoning can effectively help decision makers identify high-risk areas. It offers a cost-effective method for generating a susceptibility map and allows planners to implement preventive measures to minimize landslide impacts. Nevertheless, it is vital to recognize that the model's accuracy depends on the quality and quantity of the data used in its development. Accurate and up-to-date data are essential for constructing a reliable model, which should then be validated by comparing its results with actual landslide occurrences to ensure effectiveness. Despite the successful application of the logistic regression model for identifying landslide-prone areas along National Highway 58 in India, the study has some limitations. It only focused on easily accessible factors, excluding other crucial factors like soil characteristics, groundwater conditions, and human activities that might contribute to slope instability. Additionally, incorporating temporal and spatial data on rainfall, earthquake activity, and other natural phenomena that trigger landslides could further enhance the model's performance.

Future research could benefit from including additional data sources and variables, such as soil properties and land use patterns, to improve landslide susceptibility mapping

accuracy. Integrating machine learning and deep learning techniques with remote sensing and GIS could provide more detailed and accurate information on landslide susceptibility. Furthermore, advancements in remote sensing technology, such as high-resolution satellite imagery and LiDAR, can improve the quality of input data, leading to better susceptibility assessments. Collaboration between interdisciplinary teams, including geologists, engineers, remote sensing experts, and data scientists, can further enhance the development of comprehensive landslide susceptibility models. These models can consider a broader range of factors and employ more advanced techniques for hazard assessment and prediction. To facilitate decision making and prioritize mitigation efforts, it is also essential to engage with local communities and stakeholders, including policymakers, planners, and disaster management authorities. By incorporating local knowledge and understanding the social, economic, and environmental aspects of the affected areas, more effective and sustainable mitigation strategies can be devised. Finally, improving landslide susceptibility mapping is a crucial step towards reducing the risks associated with landslides in vulnerable regions like the Himalayas. Combining advanced techniques, interdisciplinary collaboration, and stakeholder engagement can lead to more accurate and effective susceptibility assessments, ultimately contributing to more sustainable land use planning and disaster risk management.

## 5. Conclusions

- The study leveraged a logistic regression technique to create a landslide susceptibility map at a regional scale, utilizing multiple data sources like satellite imagery, digital elevation models, and Google Earth's GIS tool.
- The analysis demonstrated that landslides are predominantly influenced by inherent terrain conditions, encompassing moderate to steep slopes, high drainage density, ridges, and other landforms.
- Eleven independent variables were analyzed for their role in landslide occurrences, with four found to exert a positive influence on landslides.
- The resultant landslide susceptibility zonation (LSZ) map spotlighted high-susceptibility zones mainly in central regions, which are marked by specific features such as terraces and proximity to streams.
- The LSZ map underscored the negative implications of unplanned infrastructure on the Himalayan terrain, especially during the monsoon season.
- The receiver operating characteristic (ROC) curve technique confirmed the study's accuracy, resulting in a commendable 92% prediction accuracy.
- The logistic regression model employed in this research is heralded as a beneficial tool for identifying landslide-prone regions.
- Emphasizing the integration of various independent variables and their synergies is crucial to creating an accurate landslide susceptibility map, which can aid in disaster prevention and planning.
- The insights from this study hold potential for application in other areas of India or globally, where geological and topographical parallels exist, validating the model's transferability.
- Future extensions of this study could delve into the cost–benefit analysis of diverse landslide mitigation tactics, such as slope stabilization and land use planning, particularly along National Highway 58 and analogous vulnerable regions.

**Author Contributions:** Conceptualization, M.S., R.K.U., S.K. and S.K.S.; methodology, M.S., R.K.U., S.K. and S.K.S.; software, M.S., R.K.U., N.K., A.S. and G.T.; validation, M.S., R.K.U., N.K., S.N.T. and A.S.; formal analysis, M.S., R.K.U., G.T. and G.M.; investigation, M.S., R.K.U., N.K., A.S., G.T., G.M. and B.A.J.; resources, M.S., R.K.U., G.T. and G.M.; data curation, M.S., R.K.U., G.T. and G.M.; writing—original draft preparation, M.S., R.K.U., G.T., N.K., A.S., G.M., S.K., S.K.S. and B.A.J.; writing—review and editing, M.S., R.K.U., G.T., G.M., S.K., S.K.S., P.K. and B.A.J.; visualization, M.S., R.K.U., G.T. and G.M.; supervision, M.S., S.K., S.K.S., G.M., P.K. and B.A.J. All authors have read and agreed to the published version of the manuscript.

**Funding:** This research received no external funding.

**Data Availability Statement:** The data used in this research is available from the first author on request.

**Conflicts of Interest:** The authors declare no conflict of interest.

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
