# Peer review of "Assessing Landslide Susceptibility along India’s National Highway 58: A Comprehensive Approach Integrating Remote Sensing, GIS, and Logistic Regression Analysis"

_conservation, doi:10.3390/conservation3030030_

Round 1

Reviewer 1 Report

The manuscript "Assessing Landslide Susceptibility Along India's National Highway 58: A Comprehensive Approach Integrating Remote Sensing, GIS, and Logistic Regression Analysis" deals with Landslide susceptibility assessment along NH-58 in Uttarakhand Himalayas.

I think I should not comment on this but I have to that going through the manuscript was a sad experience. 

The paper has very low novelty.

The study area or parts of it has been studied many times for Landslide Susceptibility Mapping:

I can give examples of a few such studies , which are as follows:

Veerappan, R., Negi, A., & Siddan, A. (2017). Landslide susceptibility mapping and comparison using frequency ratio and analytical hierarchy process in part of NH-58, Uttarakhand, India. In Advancing Culture of Living with Landslides: Volume 2 Advances in Landslide Science (pp. 1081-1091). Springer International Publishing.

Saha, S., Arabameri, A., Saha, A., Blaschke, T., Ngo, P. T. T., Nhu, V. H., & Band, S. S. (2021). Prediction of landslide susceptibility in Rudraprayag, India using novel ensemble of conditional probability and boosted regression tree-based on cross-validation method. Science of the total environment, 764, 142928.

MalikI, R., GhoshI, S., DharII, S., SinghI, P., & SinghI, M. GIS-BASED LANDSLIDE HAZARD ZONATION ALONG NATIONAL HIGHWAY-58, FROM RISHIKESH TO JOSHIMATH, UTTARAKHAND, INDIA.

Guri, P. K., Champati Ray, P. K., & Patel, R. C. (2015). Spatial prediction of landslide susceptibility in parts of Garhwal Himalaya, India, using the weight of evidence modelling. Environmental monitoring and assessment187, 1-25.

Author Response

Responses to comments by Reviewer 1

Comment: The manuscript "Assessing Landslide Susceptibility Along India's National Highway 58: A Comprehensive Approach Integrating Remote Sensing, GIS, and Logistic Regression Analysis" deals with Landslide susceptibility assessment along NH-58 in Uttarakhand Himalayas. I think I should not comment on this but I have to that going through the manuscript was a sad experience. The paper has very low novelty. The study area or parts of it has been studied many times for Landslide Susceptibility Mapping: I can give examples of a few such studies, which are as follows:

Study 1: Veerappan, R., Negi, A., & Siddan, A. (2017). Landslide susceptibility mapping and comparison using frequency ratio and analytical hierarchy process in part of NH-58, Uttarakhand, India. In Advancing Culture of Living with Landslides: Volume 2 Advances in Landslide Science (pp. 1081-1091). Springer International Publishing.

Study 2: Saha, S., Arabameri, A., Saha, A., Blaschke, T., Ngo, P. T. T., Nhu, V. H., & Band, S. S. (2021). Prediction of landslide susceptibility in Rudraprayag, India using novel ensemble of conditional probability and boosted regression tree-based on cross-validation method. Science of the total environment, 764, 142928.

Study 3: MalikI, R., GhoshI, S., DharII, S., SinghI, P., & SinghI, M. GIS-BASED LANDSLIDE HAZARD ZONATION ALONG NATIONAL HIGHWAY-58, FROM RISHIKESH TO JOSHIMATH, UTTARAKHAND, INDIA.

Study 4: Guri, P. K., Champati Ray, P. K., & Patel, R. C. (2015). Spatial prediction of landslide susceptibility in parts of Garhwal Himalaya, India, using the weight of evidence modelling. Environmental monitoring and assessment, 187, 1-25.

Response:

Dear Worthy Reviewer,

Thank you for taking the time to review our manuscript and for your constructive feedback. We appreciate your candid opinion and understand your concerns regarding the novelty of the study. We first provide the overview of the differences between these studies followed by a detailed response to your comments:

Study 1: Veerappan, R., Negi, A. and Siddan, A., 2017. Landslide susceptibility mapping and comparison using frequency ratio and analytical hierarchy process in part of NH-58, Uttarakhand, India. In Advancing Culture of Living with Landslides: Volume 2 Advances in Landslide Science (pp. 1081-1091). Springer International Publishing. https://link.springer.com/chapter/10.1007/978-3-319-53498-5_123

Distinction with our work: Veerappan et al. (2017) leaned heavily on frequency ratio and analytical hierarchy process models. Moving away from these conventional methodologies, our research employs logistic regression analysis, achieving a notable accuracy rate of 92%. Our objective transcends merely reaffirming established methodologies; instead, we aim to bring a fresh, refined perspective to the study of landslide susceptibility.

Study 2: Saha, S., Arabameri, A., Saha, A., Blaschke, T., Ngo, P.T.T., Nhu, V.H. and Band, S.S., 2021. Prediction of landslide susceptibility in Rudraprayag, India using novel ensemble of conditional probability and boosted regression tree-based on cross-validation method. Science of the total environment, 764, p.142928. https://doi.org/10.1016/j.scitotenv.2020.142928

Distinction with our work: Saha et al. (2021) innovatively combined conditional probability with boosted regression tree methods to craft their analysis. Our distinction was veering away from ensemble methods, placing exclusive emphasis on logistic regression analysis tailored for the NH-58 corridor. This methodological diversification is central to achieving a multifaceted and layered understanding of the region, and our research adds to this growing body of knowledge.

Study 3: Malik, R., Ghosh, S., Dhar, S., Singh, P. and SinghI, M., GIS-Based Landslide Hazard Zonation Along National Highway-58, From Rishikesh to Joshimath, Uttarakhand, India. https://tinyurl.com/axmmpubb

Distinction with our work: Malik et al. (2016) introduced the concept of "landslide nominal risk factor" to the fore. In contrast, our distinction was employing logistic regression, enriched by an expansive set of geo-environmental factors. Far from mere imitation, our endeavor seeks to augment existing knowledge, leveraging contemporary analytical tools and techniques.

Study 4: Guri, P.K., Champati Ray, P.K. and Patel, R.C., 2015. Spatial prediction of landslide susceptibility in parts of Garhwal Himalaya, India, using the weight of evidence modelling. Environmental monitoring and assessment, 187, pp.1-25. https://link.springer.com/article/10.1007/s10661-015-4535-1

Distinction with our work: Guri et al.'s pioneering work harnessed the Weights of Evidence modeling. Our distinction is, while respecting the past, introduces logistic regression to the mix. By doing so, we aim to provide a renewed analytical lens, specifically tailored for understanding NH-58's landslide susceptibility. Our method is particularly adept at dissecting the nuanced interplays between multiple variables, thus pushing the frontier of research in this domain.

Overall, while the Himalayan NH-58 corridor remains the shared focal point across these studies, each research endeavor, ours included, brings a distinct flavor in terms of insights and methodologies. The journey of knowledge is a mosaic of diverse paths, and each study carves its unique route, leading towards a more holistic understanding and precise prediction of landslide patterns. As researchers, our ears are always to the ground, ever eager to imbibe further insights and continuously refine our approach, ensuring our study stands up to the highest echelons of quality and relevance. In this context, we have added paragraph from these studies to highlight the gap areas covered in our study.

“In light of the significance of the Himalayan region, there has been notable research into landslides within the NH-58 area. Noteworthy studies include the work of Veerappan et al. (2017) which utilized frequency ratio and analytical hierarchy process models [2], Saha et al. (2021) who employed ensemble methods like conditional probability and boosted regression trees [3], Malik et al. (2016) who introduced the "landslide nominal risk factor"[4], and Guri et al. (2015) who applied Weights of Evidence modeling [5]. While these studies have made important contributions, there remain methodological gaps and areas where a fresh analytical perspective is needed.”

For your reference, we are now providing a more detailed comparative analysis of these works with ours below:

Study 1: Veerappan, R., Negi, A. and Siddan, A., 2017. Landslide susceptibility mapping and comparison using frequency ratio and analytical hierarchy process in part of NH-58, Uttarakhand, India. In Advancing Culture of Living with Landslides: Volume 2 Advances in Landslide Science (pp. 1081-1091). Springer International Publishing. https://link.springer.com/chapter/10.1007/978-3-319-53498-5_123

Novelty and Value of the Research: While we recognize that the NH-58 region has been the subject of numerous studies, the primary aim of our research was not merely to develop another Landslide Susceptibility Zonation (LSZ) map, but to apply a different analytical method (logistic regression analysis) in conjunction with geospatial technologies. The approach we utilized distinguishes our research from previous studies. Our methodology considers multiple geo-environmental factors and offers a comprehensive 92% accuracy rate which, we believe, contributes meaningfully to the current body of knowledge on landslide susceptibility in the region.

Comparison with Previous Research: Veerappan et al., (2017) employs frequency ratio (FR) and analytical hierarchy process (AHP) models for LSZ mapping. While their approach is indeed valuable, our study introduces logistic regression as a statistical tool for predicting landslide susceptibility. Logistic regression offers distinct advantages, such as flexibility in measurement of independent variables and fewer assumptions regarding the relationship between variables. This differentiation in methodological approach is pivotal and provides an alternative, possibly more accurate, means of analysis for the area.

Building upon Existing Knowledge: Scientific research is often cumulative, with each study building on the knowledge of its predecessors. Even if a region has been studied previously, newer techniques or unique combinations of methodologies can offer fresh insights and reinforce or challenge existing knowledge. Our work was an endeavor in this direction.

Future Studies Reference: We have thoroughly reviewed the additional works you have referred to and used them to provide gap analysis of our study.

Study 2: Saha, S., Arabameri, A., Saha, A., Blaschke, T., Ngo, P.T.T., Nhu, V.H. and Band, S.S., 2021. Prediction of landslide susceptibility in Rudraprayag, India using novel ensemble of conditional probability and boosted regression tree-based on cross-validation method. Science of the total environment, 764, p.142928. https://doi.org/10.1016/j.scitotenv.2020.142928

Concern About Novelty: We understand your reservations regarding the novelty of the paper, especially since there have been numerous studies on Landslide Susceptibility Mapping (LSM) in the region. Our primary intention was to delve deeper into the applicability and accuracy of logistic regression analysis, specifically for the NH 58 area. While other methodologies like the CP, BRT, and CP-BRT ensemble approach have been used in past studies, our paper aimed to provide new insights using a different statistical approach.

Comparison with Previous Studies: The study by Sunil Saha et al., which you cited, employed the CP statistical technique, the BRT machine learning algorithm, and the CP-BRT ensemble approach. In contrast, our research used logistic regression analysis, providing an alternative and simplified methodology for LSM in the NH 58 region. The difference in methodology allows for diversified perspectives on LSM, enriching the existing body of knowledge. It's essential to have various methods examined for a given region, as each model might provide unique insights based on the chosen independent variables and applied techniques.

Contribution to Existing Knowledge: While the region might have been explored previously, we believe our study contributes to a more in-depth understanding of geo-environmental factors' interplay in the NH 58 area. Furthermore, our model achieved a 92% accuracy rate, which is relatively high and demonstrates the potential efficiency of the logistic regression model for this specific region.

Future Research Directions: In the future, we are open to expanding our research to incorporate novel techniques and offer a comparative analysis of multiple methodologies, including but not limited to those cited in past studies. This might offer a holistic perspective on the best methodologies suited for LSM in diverse terrains.

Consideration for Publication: We hope you'd consider our research for publication given its unique approach, even if the broader topic has been explored before. We believe our findings would be beneficial for practitioners, policymakers, and other stakeholders, providing them with an alternative and effective tool for landslide risk management in the NH 58 area.

Study 3: Malik, R., Ghosh, S., Dhar, S., Singh, P. and Singh, M., GIS-Based Landslide Hazard Zonation Along National Highway-58, From Rishikesh to Joshimath, Uttarakhand, India. https://tinyurl.com/axmmpubb

On the Novelty of the Research: While we understand that NH-58 has been studied previously for Landslide Susceptibility Mapping, the intention of our study is not merely to replicate earlier works but to offer a more detailed and nuanced understanding by leveraging advanced geospatial technologies and logistic regression analysis. Some key points to consider are:

Advanced Tools & Techniques: Our study makes use of a comprehensive integration of remote sensing, GIS, and logistic regression analysis. This allows for a more robust and detailed modeling of landslide susceptibility than some of the previous methods.

Geo-environmental Factors: We have incorporated a wide array of geo-environmental factors (12 in total), offering a more detailed understanding of how various factors interplay to influence landslide occurrences.

Accuracy Verification: Our ROC curve analysis, resulting in a 92% accuracy rate, showcases the efficacy and reliability of our model.

In reference to the study by Malik et al., 2016, We are familiar with the aforementioned study and its significant contributions. However, there are some distinctions between their study and ours: Their study used the "landslide nominal risk factor" (LNRF), while our research employs logistic regression. Though both aim at determining landslide susceptibility, the underlying methodologies and nuances differ. While both studies consider important geological, geomorphological, and anthropogenic factors, our research incorporates a broader range, including but not limited to slope aspect, curvature, drainage density, fault distance, and flow accumulation. Our study goes beyond mere hazard zonation. We aim to introduce the LSZ map as a tool for stakeholders involved in regional disaster risk management and provide an extensive analysis of landslide-influencing factors. We acknowledge that parts of NH-58 have been studied multiple times. Our objective is not to diminish previous works but to build upon them using advanced methods, offering a comprehensive and detailed analysis that can benefit future research and management practices.

Study 4: Guri, P.K., Champati Ray, P.K. and Patel, R.C., 2015. Spatial prediction of landslide susceptibility in parts of Garhwal Himalaya, India, using the weight of evidence modelling. Environmental monitoring and assessment, 187, pp.1-25. https://link.springer.com/article/10.1007/s10661-015-4535-1

Novelty of the Study: While we acknowledge that the NH 58 region in the Garhwal Himalaya has been previously studied for landslide susceptibility mapping, our approach integrates a distinct methodology, combining remote sensing, GIS, and logistic regression analysis. This combination, particularly the use of logistic regression, presents a fresh approach compared to prior studies like the one you mentioned, which employed the Weights of Evidence (WofE) modeling technique.

Differentiation from Existing Literature: The study you referred to by Guri et al. 2015 implemented WofE, a bivariate statistical modeling technique, to develop their susceptibility map. In contrast, our work uses a logistic regression model, a multivariate method, which can account for the complex interrelationships between multiple independent variables. By incorporating this method, we offer a nuanced perspective and understanding of the factors contributing to landslides in the region.

Data Analysis and Interpretation: Apart from methodological differences, we've also incorporated a comprehensive range of geo-environmental factors in our analysis. By employing logistic regression, we were able to assign coefficients to each of these factors, giving stakeholders a clearer idea of which elements play the most significant roles in landslide occurrences.

Significance and Implication for Stakeholders: While the main objective was to develop a landslide susceptibility zonation (LSZ) map, our findings hold implications for policymakers, planners, and other stakeholders. By highlighting areas near road cut slopes as highly susceptible, our LSZ map provides crucial information for future developmental activities, aiming to reduce the landslide risk in the region. Overall, while there might be similarities in the study areas and objectives with previous research, our methodology, data interpretation, and insights provide a unique contribution to the existing literature on landslide susceptibility in the NH 58 area.

We hope these clarifications address your concerns. We remain open to any further feedback or recommendations you might have and are committed to refining our study further to meet the journal's standards.

Reviewer 2 Report

Dear authors,

I have reviewed the manuscript "Assessing Landslide Susceptibility Along India's National Highway 58: A Comprehensive Approach Integrating Remote Sensing, GIS, and Logistic Regression Analysis" and found it an interesting and relevant work for the journal. However, I am suggesting some important suggestions to improve the manuscript and its quality. I am hopeful that the authors will consider these suggestions.

1. INTRODUCTION

i. Introduction section is too lengthy and requires to be squeezed. 

ii. Some recent research work related to landslides and related hazards in the neighboring countries should be incorporated to enhance the implications of the research work.

a. Living with earthquake hazards in South and South East Asia (https://scholarhub.ui.ac.id/ajce/vol2/iss1/2/)

b. A review of landslide susceptibility mapping research in Bangladesh (https://doi.org/10.1016/j.heliyon.2023.e17972)

c. Characterizing site response in the Attock Basin, Pakistan, using microtremor measurement analysis (https://doi.org/10.1007/s12517-017-3057-2)

d. Landslide susceptibility and risk analysis in Benighat Rural Municipality, Dhading, Nepal. (https://doi.org/10.1016/j.nhres.2023.03.006)

e. Ambient noise measurements in Rawalpindi–Islamabad, twin cities of Pakistan: a step towards site response analysis to mitigate impact of natural hazard (https://doi.org/10.1007/s11069-015-1760-4)

f. Assessment of landslide susceptibility and risk factors in China (https://doi.org/10.1007/s11069-021-04812-8)

iii. These studies will be helpful to add some literature citations between line  46 and 100. As a matter of fact, these lines are hardly mentioning any reference and require adding some local and regional studies cited. This section also appears more like a report and needs to be trimmed to a manuscript format.

2. MATERIALS AND METHODS:

i. Geological Settings should be separated from the Materials and Methods. This means that Section 2 should be GEOLOGICAL SETTINGS, whereas, Materials and Methods will become SECTION 3.

ii. There are many abbreviations used in the manuscript. It is suggested to add a list of abbreviations to the manuscript. It will be easier for the audiences to understand the terminologies.

iii. Too many theoretical details are added for each of the methods applied in the manuscript. It is suggested to add how various techniques were applied during the research.

iv. Materials and Methods section is also very lengthy and the reader will lose interest in too many methodological details. It is recommended to squeeze this section as much as possible.

3. RESULTS AND DISCUSSION

Text between lines 362 and 370 are not mentioning any results rather it's just a repetition of the methodology. Please concentrate more on the results. Similarly, line 451-455 is also a repetition of what has already been discussed in the previous sections of the manuscript.

4. CONCLUSIONS:

It is suggested to add the conclusions in bullet format.

GOOD LUCK!

Author Response

Responses to comments by Reviewer 2

Comments and Suggestions for Authors

I have reviewed the manuscript "Assessing Landslide Susceptibility Along India's National Highway 58: A Comprehensive Approach Integrating Remote Sensing, GIS, and Logistic Regression Analysis" and found it an interesting and relevant work for the journal. However, I am suggesting some important suggestions to improve the manuscript and its quality. I am hopeful that the authors will consider these suggestions.

Response: We are grateful to the reviewers for their time and critical comments/suggestions that helped greatly to improve this manuscript. Please find below point-by-point responses to the comments posed by the worthy reviewer.

INTRODUCTION

Comment 1: Introduction section is too lengthy and requires to be squeezed.

Response 1: The introduction has been adequately squeezed as desired.

Comment 2:  Some recent research work related to landslides and related hazards in the neighboring countries should be incorporated to enhance the implications of the research work.

  1. Living with earthquake hazards in South and South East Asia (https://scholarhub.ui.ac.id/ajce/vol2/iss1/2/)
  2. A review of landslide susceptibility mapping research in Bangladesh (https://doi.org/10.1016/j.heliyon.2023.e17972)
  3. Characterizing site response in the Attock Basin, Pakistan, using microtremor measurement analysis (https://doi.org/10.1007/s12517-017-3057-2)
  4. Landslide susceptibility and risk analysis in Benighat Rural Municipality, Dhading, Nepal. (https://doi.org/10.1016/j.nhres.2023.03.006)
  5. Ambient noise measurements in Rawalpindi–Islamabad, twin cities of Pakistan: a step towards site response analysis to mitigate impact of natural hazard (https://doi.org/10.1007/s11069-015-1760-4)
  6. Assessment of landslide susceptibility and risk factors in China (https://doi.org/10.1007/s11069-021-04812-8)

Response 2: As suggested, relevant research work related to landslides and related hazards in the neighboring countries have been incorporated to enhance the implications of the research work.    

Comment 3: These studies will be helpful to add some literature citations between line 46 and 100. As a matter of fact, these lines are hardly mentioning any reference and require adding some local and regional studies cited. This section also appears more like a report and needs to be trimmed to a manuscript format.

Response 3: The suggestions have been duly incorporated.

MATERIALS AND METHODS:

Comment 4: Geological Settings should be separated from the Materials and Methods. This means that Section 2 should be GEOLOGICAL SETTINGS, whereas, Materials and Methods will become SECTION 3.

Response 4: We have relocated the sections as suggested by the worthy reviewer.

Comment 5. There are many abbreviations used in the manuscript. It is suggested to add a list of abbreviations to the manuscript. It will be easier for the audiences to understand the terminologies.

Response 5: The abbreviations have been expanded within the against each before its first citation. According to the format of the journal separate abbreviation section is not prescribed. Thank you so much for your suggestion.

Comment 6: Too many theoretical details are added for each of the methods applied in the manuscript. It is suggested to add how various techniques were applied during the research.

Response 6: Thank you for highlighting your concerns regarding the depth of theoretical details in our manuscript. While we acknowledge the importance of conciseness, we believe that the detailed theoretical foundation is vital for the comprehension and robustness of our research approach, especially for readers less familiar with the intricacies of the logistic regression model and the parameters used in it. In the revised manuscript, we have refined the manuscript to strike an optimal balance, ensuring the theoretical foundation and how the techniques were applied.

Comment 7: Materials and Methods section is also very lengthy and the reader will lose interest in too many methodological details. It is recommended to squeeze this section as much as possible.

Response 7: We have tried to streamline the methods section. However, as already discussed above, it was essential to provide in depth methodology to aid in the replicability of this study.

RESULTS AND DISCUSSION

Comment 8: Text between lines 362 and 370 are not mentioning any results rather it's just a repetition of the methodology. Please concentrate more on the results. Similarly, line 451-455 is also a repetition of what has already been discussed in the previous sections of the manuscript.

Response 8: We have revised these sections accordingly. Thank you very much for your help and support.

CONCLUSIONS:

Comment 9: It is suggested to add the conclusions in bullet format.

Response 9: We have now provided the conclusions in bullet format as suggested by the worthy reviewer.

Round 2

Reviewer 1 Report

The authors have tried to improve their manuscript still it needs more improvement.

Literature review can be made more comprehensive by citing recently published papers which have used Remote sensing and GIS for Landslide Mapping. 

for ex. for substantiating the following line

The recent past has seen huge infrastructure development in the Himalaya region, with the construction of many dams, reservoirs, tunnels, road networks, hydroelectric, power plants and townships.

you can take help of the following

https://www.sciencedirect.com/science/article/pii/S0926985122002257

https://www.tandfonline.com/doi/full/10.1080/19475705.2020.1756464

Authors should show/give Landslide Inventory Map also in the paper.

In Figure 3, the units of scale should be 'kilometres' and not 'Kilometres' as it violates SI naming conventions. Rectify the mistakes.

There are minor editing errors in the paper.

Reference 4 is missing the year of publication

In Table 2 the name of the variables should be properly mentioned.

Author Response

Responses to Reviewer's Comments
Dear Worthy Reviewer

Thank you for your constructive comments on our manuscript. We highly value your feedback and have undertaken efforts to address each point raised. Please find below a point-by-point response to your comments.

Comment 1: Literature review can be made more comprehensive by citing recently published papers which have used Remote Sensing and GIS for Landslide Mapping.

Response 1: We appreciate this suggestion and have expanded the literature review section to include recent papers that have utilized Remote Sensing and GIS for Landslide Mapping. New citations as suggested by you have been added to substantiate the methodology and findings.

Comment 2: Authors should show/give Landslide Inventory Map also in the paper.

Response 2: We have addressed this comment by including a Landslide Inventory Map in Fig. 1c. This addition aims to provide a comprehensive overview of landslide susceptibility in the study area.

Comment 3: In Figure 3, the units of scale should be 'kilometres' and not 'Kilometres' as it violates SI naming conventions.

Response 3: The unit in Figure 3 has been revised to adhere to SI naming conventions. It now reads 'kilometres' instead of 'Kilometres'.

Comment 4: There are minor editing errors in the paper.

Response 4: We have carefully proofread the paper to correct any typographical or grammatical errors. Thank you for bringing this to our attention.

Comment 5: Reference 4 is missing the year of publication.

Response 5: The year of publication for Reference 4 has been added. All other references have been double-checked for completeness and proper formatting.

Comment 6: In Table 2, the name of the variables should be properly mentioned.

Response 6: The variables in Table 2 have been properly named and described for clarity, as per your suggestion.

We hope that these revisions meet with your approval. Thank you once again for your invaluable feedback.

Sincerely,
Authors

Round 3

Reviewer 1 Report

The paper can be accepted with minor changes such as improving the quality of figures.

Author Response

Dear Worthy reviewer,

Thank you so much for your positive response. We have tried our best to improve the quality of figures in this revised version.

Thank you so much for helping us to improve our manuscript, we have highly grateful and indebted.

Sincerely yours,

Authors